# Maternal Expectations and Overinvolvement, and Child Emotion Regulation in Children with and Without Anxiety Disorders: An Experimental Observational Study

**DOI:** 10.3390/ijerph22121802

**Published:** 2025-11-28

**Authors:** Eva-Maria Fassot, Brunna Tuschen-Caffier, Vera Hauffe, Julia Asbrand

**Affiliations:** 1Department of Clinical Psychology and Psychotherapy, Institute for Psychology, University of Freiburg, 79085 Freiburg im Breisgau, Germany; eva-maria.fassot@psychologie.uni-freiburg.de (E.-M.F.); tuschen@psychologie.uni-freiburg.de (B.T.-C.); vera.hauffe@psychologie.uni-freiburg.de (V.H.); 2Department of Clinical Psychology of Childhood and Adolescence, Institute for Psychology, University of Jena, Semmelweisstr. 12, 07743 Jena, Germany

**Keywords:** emotion regulation, anxiety, expectation, children, overinvolvement, family

## Abstract

This study explores how child emotion regulation (ER) and maternal overinvolvement contribute to the maintenance of anxiety disorders (ADs) in children. Building on the tripartite model on the impact of the family on children’s emotion regulation and adjustment, it experimentally examines the impact of maternal expectations on overinvolvement and child distress. 65 children with ADs (ages 8–15) and 64 healthy controls (HCs) and their mothers participated in a tangram task, with manipulated maternal expectations. Mother–child interactions were observed for maternal involvement and child behavior (ER, distress), with children reporting their emotional reactivity. Against expectations, maternal involvement did not differ between groups and was not influenced by expectations. HC children had better ER abilities than those with ADs. Emotional reactivity moderated the relationship between overinvolvement and ER problems. Children with ADs exhibited more distress, unaffected by maternal expectations. Further research is needed to clarify the relationship between parental behavior and child behavior, particularly regarding emotional reactivity.

## 1. Introduction

Anxiety disorders (ADs) are among the most prevalent mental disorders in children [1,2,3]. They have significant adverse effects on both children and families [1,4,5]. These disorders also predict mental health issues in adulthood [6,7] and impose substantial costs on families and society [8]. The COVID-19 pandemic further exacerbated anxiety levels in children [9,10]. Understanding the factors contributing to the maintenance of ADs is essential for developing theoretical models of their development and persistence. One important child factor in this context is emotion regulation (ER), which is considered an underlying transdiagnostic factor for ADs [11,12]. ER is defined as “extrinsic and intrinsic processes responsible for monitoring, evaluating, and modifying emotional reactions, especially their intensity and duration, to achieve one’s goals” [13] (pp. 27–28).

Many studies have suggested that dysfunctional ER is an important factor in the maintenance of ADs in children [12,14,15,16,17,18,19]. In their meta-analysis, Schäfer et al. [16] identified an association between anxiety symptoms and a less frequent use of adaptive ER strategies and a more frequent use of maladaptive strategies. The strongest negative associations were found with acceptance and the strongest positive associations with avoidance and rumination. However, a majority of studies have relied on self-report measures or parent-reported assessments of children’s ER, which could limit the generalizability of the finding [16] such measures can be influenced by cognitive abilities, daily mood, and social desirability effects [20]. To overcome these limitations, we used an observational system to directly evaluate ER in children with and without ADs.

The maintenance of both ADs and ER is thought to be influenced by parental factors, such as overinvolvement [21,22,23,24,25]. Overinvolvement “is characterized by excessive interfering in a child’s behavior, thoughts, and feelings, and encouragement of excessive dependence on the parent” ([26], p. 18) and is a broader dimension integrating subcomponents like overcontrol, overprotection, intrusiveness [27,28]. Overinvolvement is a well-documented construct in anxiety disorder research, frequently assessed via observational methods (e.g., [22,27]) that include measurable behavioral indicators. One model integrating these different factors (AD, ER and overinvolvement) and describing the possible influence of the different aspects is Morris et al.’s [29] tripartite model of the family. Child ER is the central factor of this model. According to the model, overinvolvement is considered as part of the emotional climate within the family and directly influences both child ER and psychopathology such as ADs. Additionally, ER is considered as a mediator between overinvolvement and ADs. Several findings support Morris et al.’s [29] assumption that overinvolvement is relevant to both the maintenance and the development of ER and ADs [11,22,24,30,31] and also for to the interplay between overinvolvement and ADs, with ER serving as a mediator [31,32,33]. Also, Allen et al. [34] emphasized in their model of youth anxiety incorporating the developmental perspective that parental control is a key factor—particularly during middle childhood and among children with behavioral inhibition (BI) vulnerability—in the development of emotion regulation difficulties, a precursor to ADs. This aligns with Rapee et al.’s [35] findings that parental overcontrol exerts significant influence on ADs during middle childhood and the onset of adolescence. However, it remains unclear whether overinvolvement affects child ER in immediate interactions or primarily over longer developmental periods—particularly during middle childhood and the transition to early adolescence. Situational processes in parent–child interactions could provide insights into the mechanisms underlying long-term effects, but empirical evidence is lacking. More overinvolvement and less functional ER might contribute to the maintenance of ADs in the long term.

Both Allen et al. [34] and Morris et al. [29] pointed out the influence of child factors such as emotional reactivity on the relationship among parenting behavior, ER, and psychopathology. In the model of Morris et al. [29], emotional reactivity is considered a moderator of the relationship between parental overinvolvement and ER and the child adjustment. Also, the current literature emphasizes the distinction between ER and emotional reactivity [14,19,36,37]. Children with ADs tend to show higher emotional reactivity compared to HC children [14,19,38]. The interaction between parenting behavior and emotional reactivity has been a subject of interest in psychological research [39,40]. In an experiment with a retrospective design, emotional reactivity served as a moderator between maternal psychological control and internalizing problems in children [41]. This aligns with the model, which describes the long-term effects of these interactions. It is crucial to investigate if this interaction can also be observed to have a direct effect on ER in a situational analysis rather than a retrospective design. This could serve as the foundational explanation for long-term effects. Utilizing the tripartite model of family influence proposed by Morris et al. [29] as a theoretical background, we expected that overinvolvement would predict dysfunctional ER in children and that this relation would be moderated by the emotional reactivity of the children being stronger for children with higher emotional reactivity.

For the direct influence of overinvolvement on the maintenance of ADs, two meta-analyses revealed that parents of children with ADs exhibited more parental control [42,43]. The effect was stronger in studies using observational measures compared to self-report measures as well as in studies comparing healthy control (HC) children to clinical children vs. subclinical samples; [42]. For instance, Hudson and Rapee [22] found with an observational design that parents of children aged 7–12 years with ADs displayed more overinvolvement compared to parents of HC children.

The underlying reasons for parents’ overinvolvement remain relatively unexplored. One plausible explanation posits that parents of children with ADs may expect their children to experience higher stress and cope less effectively with distress than parents of healthy children [44,45]. To test this assumption, Creswell et al. [46] conducted an experimental study wherein they manipulated parents’ expectations about their healthy children’s abilities. Parents were presented with a difficult anagram task and were given instructions that induced either positive expectations or negative expectations that their child would struggle with the task. Parents who received the negative instructions exhibited more overinvolvement, supporting the notion that parents seek to avoid their child experiencing distress. However, this experimental design has not been extended to comparisons of children with ADs and HC children. Parental expectation is thought to be especially important for overinvolvement of parents of children with ADs [44,45]. Thus, our study aimed to investigate this effect in mothers of children with ADs compared to mothers of HC children. In Creswell et al.’s [46] study, the influence of instructions on child behavior and affect was also investigated. However, no significant differences were found in child behavior or mood. This pattern might differ in clinical samples, where stronger effects are likely to be observed [46,47]. Given the weak theoretical background for this hypothesis, we explored the effect of instructions on child behavior in an exploratory manner. In summary, we wanted to clarify the influence of child factors (i.e., ER, emotional reactivity and child distress) and the role of maternal factors such as overinvolvement and expectations—and their interplay—in the maintenance of ADs in children, building on the tripartite model of family influence of Morris et al. [29] as a theoretical background.

### The Current Study

This study employed a quasi-experimental between-subjects design to examine the effects of maternal expectations and group membership (AD vs. HC) on ER and maternal overinvolvement. To ensure comprehensive assessment, both observational and self-report measures were employed to capture multidimensional aspects of child and maternal behavior. We expected children with ADs to show more observed dysfunctional ER than HC children (Hypothesis 1). Additionally, we expected mothers of children with ADs to show more overinvolvement than mothers of the HC children in an experimentally induced frustrating situation (Hypothesis 2). We expected an interaction effect of group and mothers’ expectations and that overinvolvement would be strengthened by negative expectations, especially for mothers of the AD group (Hypothesis 2).

We expected the level of maternal involvement to predict the level of emotional dysregulation in children, and the association between maternal involvement and problems in ER to be moderated by the emotional reactivity of the children. We expected that the association would be stronger for children with higher emotional reactivity (Hypothesis 3). We tested for this in the entire sample, regardless of group membership. Additionally, we wanted to investigate the influence of instructions on the behavior of the child—especially on child distress—in an exploratory fashion.

## 2. Materials and Methods

### 2.1. Participants

According to an a priori analysis with G*Power version 3.1.9.7 for Hypotheses 1 and 2 (Faul et al., [48]; analysis of variance [ANOVA], fixed effect, main effects, and interaction, effect size *f* = 0.25 [23], two groups, *df* = 1, α = 0.05, β = 0.8), we needed 128 participants. For the moderated regression analysis (Hypothesis 3) with three predictors on the outcome variable, considering an effect size of *ΔR*^2^ = 0.25 [41] we needed a sample size of *n* = 77.

The sample consisted of *N* = 129 mother–child dyads with children age 8–15 years. The AD group comprised *n* = 65 participants with a primary diagnosis of any AD (social anxiety disorder: *n* = 29, 45%; specific phobia: *n* = 20, 31%; generalized anxiety disorder: *n* = 12, 18%; separation anxiety: *n* = 4, 6%). Children who were undergoing treatment or taking medication for their AD were excluded. Comorbidities were allowed and were present in *n* = 32 (49%) of *n* = 65 participants. The healthy control (HC) group comprised *n* = 64 mother–child dyads without any lifetime diagnoses of the children. Table 1 shows participant characteristics for the two groups. The groups did not differ in any sociodemographic variable. As expected, there were differences related to child psychopathology. The AD group showed significantly more symptoms compared to the HC group (see Table 1). Data were collected between 2016 and 2022; *n* = 81 mother–child dyads participated during or after the COVID-19 pandemic. The pre-pandemic HC group (*n* = 29) and the HC group that participated during or after the pandemic (*n* = 35) showed no statistically significant differences in psychopathological measures (Spence Children’s Anxiety Scale [SCAS] mother report, SCAS child report, Symptom Checklist 27 [SCL-27], Child Depression Inventory [CDI]). Likewise, the pre-pandemic AD group (*n* = 19) and AD group participating during or after the pandemic (*n* = 46) did not exhibit significant variations in psychopathological outcomes according to standardized measures. The local ethics committee approved the study. Prior to participation, written informed consent was obtained from both the children and their parents. The study was preregistered to ensure transparency and reproducibility. The preregistration can be accessed at: https://aspredicted.org/18B_ZYY (accessed on 5 May 2023)

### 2.2. Procedure

This study was conducted in Germany at [location excluded to ensure authors’ anonymity]. Participants were recruited through flyers distributed in schools, medical offices, holiday day letters to parents and children through schools or directly. Families interested in participating contacted the study center. A brief telephone interview screened potential participants, who were then invited to a diagnostic session. Only mother child dyads of healthy children or children with a potential AD were invited. We used an individual structured clinical interview (Kinder-DIPS) [53] to validate diagnoses. This involved separate discussions with parents and children, assessed by trained and supervised interviewers. To ensure the quality of the diagnosis, a weekly supervision session was conducted with a practicing therapist. The Kinder-DIPS interview has been established as a reliable and valid assessment instrument [54].

Additionally, mothers and children above the age of 10 years completed various questionnaires through an online platform. Only children older than 10 years received the questionnaires, as the questionnaires were standardized only from this age onwards, and because reading skills are sufficiently developed at this age. Mother–child dyads were informed that the purpose of the study was to explore how children solve difficult tasks. The laboratory session began with a baseline measurement, followed by a 10 min computer puzzle game against the experimenter. To induce slight frustration and set the stage for the subsequent instructions given to the mothers, children won in the first 5 min, but in the last 5 min, the experimenter caught up and ultimately won the game. After a short break, the experimental manipulation was performed. Mothers either were informed that we believed on the basis of our observation of the first game that their child would struggle in the upcoming puzzle task (negative expectation) or were given no specific information (neutral expectation; *n* = 32 randomized mothers of the HD group and *n* = 32 of the AD group got the neutral instruction and *n* = 32 mothers of the HC group and *n* = 33 mothers of the AD group received a negative instruction). Children were tasked with solving as many as 12 tangram puzzles within a 10 min period, with a later evaluation promised [22]. A tangram is a Chinese puzzle with seven geometric pieces cut from a square. The goal is to form specific shapes using all pieces. The experimental setup simulated tackling difficult problems together, such as homework, for children and mothers and aimed to induce mental stress and frustration. In the neutral condition, mothers got similar instructions to Hudson and Rapee [22]: “In the next block, your child’s task will be to solve a puzzle. This task is similar to the one on the computer, but it will not be performed on the computer. You can help your child, if you think he/she really needs it”.

In the negative condition the following instruction was added Creswell et al., [46]:

“The tasks are very challenging. Based on the previous task, we assume that your child will have difficulty solving the puzzle. This may become frustrating for your child as the task progresses”.

Mother–child interactions were video-recorded for a total duration of 10 min per dyad. For coding purposes, the first 5 min of each interaction were selected and systematically analyzed using standardized behavioral coding systems. This timeframe was chosen to capture the initial dynamic of the interaction when maternal and child behaviors are most responsive to the task demands, and ensure feasibility for reliable coding while maintaining ecological validity. The duration for the mother–child Tangram interaction task was selected to align with established protocols in prior research. Specifically, this timeframe corresponds to the duration used in studies by Hagstrøm et al. [55].

Later, seven trained coders (2 or 3 independent raters for each coding system), blind to the child’s diagnosis and experimental condition, rated the videos. Each coder was trained with eight videos; after the first 10 videos, the raters could discuss differences to make future ratings more precise.

During the laboratory session, mothers and children repeatedly reported on their emotional experiences (see Figure 1). After the last emotion rating, mothers and children attended a questionnaire session to discuss evaluate the performance of the child during the computer game. After the debriefing, participants received a gift voucher with a value of 35 euros. Children with an AD were also offered the opportunity for treatment at the associated outpatient clinic.

### 2.3. Materials

#### 2.3.1. Psychometric Measures

The Spence Children’s Anxiety Scale (SCAS) [50,51]. The SCAS measures anxiety in children and adolescents between 8 and 17 years of age on a 4-point Likert scale from never to always. It contains a total score with a range of 0 to 114 and six subscales: panic attack and agoraphobia, separation anxiety disorder, obsessive compulsive disorder, social phobia, generalized anxiety disorder, and physical injury fears. Parent and child versions were applied. The German version has shown good psychometrical properties [56]. In the current sample, the internal consistency was excellent for both versions (αMother = 0.91, αChild = 0.94).

The Child Depression Inventory (CDI) [49]. The CDI consists of 27 items and measures symptoms of dysthymia and depression rated by children and adolescents between 8 and 16 years of age. The German CDI [57] was shown to have an excellent internal consistency (α = 0.92). The internal consistency of the CDI in the current sample was good (α = 0.88).

Symptom Checklist 27 (SCL-27) [52]. We used the SCL-27 to screen psychopathological symptoms in participating mothers. It consists of six subscales measuring depressive, dysthymic, vegetative, agoraphobic, sociophobic, and mistrust symptoms. The internal consistency of the sum score was good (α = 0.88).

#### 2.3.2. Measures During the Laboratory Session

During the laboratory session, the children and mothers rated their emotions at five points (see Figure 1) on the Scales for Iconic Self-Assessment of Anxiety in Children (ISAAC; [58]; Permission to use was obtained from Silvia Schneider, University of Bochum, Germany). They rated the degree of anxiety, sadness, anger, disappointment, shame, happiness, relaxation, and pride using a visual analog scale with endpoints not at all and very intense. Children were presented with various facial expressions depicting different emotions for orientation purposes. To measure emotional reactivity during the tangram puzzle in children, we used the rating of the children after the tangram. As we wanted to induce negative emotions, the negative emotions (anxiety, sadness, anger, disappointment, shame) were added to an overall score of negative emotions. The ISAAC has previously been used to measure situational emotionality [58]. Also, Carthy et al. [59] and Herres et al. [38] used the subjective emotion rating as an index of emotional reactivity. The internal consistency for overall score of negative emotions of the children measured after the tangram puzzle was acceptable (α = 0.79). An exploratory factor analysis of the ISAAC of the children measured after the tangram puzzle revealed a two-factor structure (see Appendix A, Table A1). Negative emotions consistently loaded positively on one factor, and positive emotions loaded on the other. At the end of the session, mothers and children received a questionnaire to judge the performance of the child during the computer game.

#### 2.3.3. Observation Measures During the Tangram

The Tangram Emotion Coding Manual (TEC-M) [55]. (Permission to use and adapt the coding manual was obtained from Julie Hagstrøm, Region Hovedstsens Psykiatri.) The TEC-M is based on the process model of emotion regulation [60]. It is an observational coding system designed to assess children’s ER profiles during parent–child interactions while solving Tangram puzzles. Developed and validated by Hagstrøm et al. [55,61], the TEC-M integrates theory-driven and data-driven items to capture observable aspects of ER processes in real-time interaction. There are eight items assessing parent behavior (overinvolvement, control, avoidance, verbal reappraisal, support/sensitivity, positive expressions, negative expressions, and tension), and eight items for child behavior (control, avoidance/resignation, narration, verbal reappraisal, reassurance-seeking behavior, aggression, positive expressions, negative expressions) measuring various ER strategies at different timepoints. Additionally, one item evaluates emotional warmth in the parent–child dyad. Finally, the overall ER scale (EmReg score) is built by integrating the overall ER ability of the child and taking into account the perceived difficulties of the child as well as parental behavior. Although the judgment of individual items is not numerically integrated, the EmReg score is influenced by prior evaluations of those items, shaping a mindset that guides subsequent judgments. It is scored on a 5-point Likert scale of 1 (very poor ER skills) to 5 (excellent ER skills). In this study, EmReg score (mean of the three raters) is used to measure the ER abilities of the child. Diverse viewpoints regarding the scale level pertaining to the construct of ER have been observed in the extant literature [55]. Describing the ER construct in a continuous manner has proven to be challenging; nevertheless, other analogous instruments measuring ER, such as the Emotion Regulation Questionnaire [62] operate under this fundamental assumption. Consequently, we posit the EmReg score as interval scaled for analytical purposes. The TEC-M is a validated instrument [63]. For this study, the English version was translated into German and checked by a bilingual speaker. Interrater reliability (ICC, 2,k; two way random effects, absolute agreement, multiple raters) based on [64] for the three coders was good (0.60).

Tangram Coding System (TCS): Mothers [22]. (Permission to use and adapt the coding manual was obtained from Jennie Hudson, Macquarie University in Sydney, Australia). The TCS gauges the parenting factors Involvement and Negativity through nine scales on a nine-point continuum (0 to 8). The first five scales measure Involvement (mean of the scales): General Degree of Involvement, Degree of Unsolicited Help, Touching of the Tangram Pieces, Mother’s Posture, and Mother’s Focus. Overinvolvement measures the degree to which the mother intervenes in the child’s task performance, including intrusive, overcontrolling, or overly supportive behaviors (e.g., taking over the task, excessive verbal directives. A higher score on this factor indicates greater maternal involvement and intrusiveness during the task. The remaining four scales measure Negativity: General Mood of the Interaction, Mother’s Degree of Positive Affect, Mother’s Tension, and Mother’s Degree of Criticism. For the current study, a previously validated German translation was used [27]. As the German version has been used previously [27] and for economy reasons, two and not three coders rated the videos. For this study, the factor Involvement (mean of the two coders) was used to measure the overinvolvement of mothers. Interrater reliability for Involvement was excellent (0.93).

TCS: Children [22]. (Permission to use and adapt the coding manual was obtained from Jennie Hudson, Macquarie University in Sydney, Australia). To explore child behavior that might be a reaction to the parent behavior, we employed a previously developed observational system [27] comprising nine scales adapted to a 9-point continuum. These scales were Child’s General Mood, Child’s Degree of Positive Affect, Child’s Tension, Nonpersistence on Task, Noncompliance with Parental Behavior, Child’s Dependence, Helplessness (verbal/nonverbal), Perceived Difficulty of the Task and nonresponsiveness to Mother’s Questions, Comments, and Behaviors. Following Asbrand et al. [27], we formed the Child Distress factor using Child’s General Mood, Child’s Degree of Positive Affect, Child’s Tension, and Noncompliance with Parental Behavior. Because we had no clear hypotheses for dependent child behavior as well as a lack of psychometric quality, we included Distress only in the following analyses. Distress refers to the observable behavior children exhibit [27] and refers to a more persistent emotional strain or tension. It differs from emotional reactivity, which describes the direct internal emotional response. Interrater reliability for Distress was good (0.70).

### 2.4. Data Analysis

All statistical analyses were calculated using IBM SPSS version 25. Additionally, for the moderation hypothesis (Hypothesis 3), we used the Hayes [63] package for SPSS. Intraclass correlations for each scale of the different observational measures were calculated following Shrout and Fleiss [64] to determine the interrater reliability of the different coders (ICC, 2,k; two way random effects, absolute agreement, multiple raters). The interpretation of the reliability coefficient values was based on the guidelines by Cicchetti [65].

For the manipulation check, mothers’ self-reported emotions before and after the tangram puzzle were assessed using mixed-model ANOVAs for each emotion of the ISAAC. Hypothesis 1 was tested with an ANOVA examining the effect of diagnosis (HC vs. AD) on observed child ER operationalized by the EmReg score of the TEC-M. Hypothesis 2 was investigated with an ANOVA considering diagnosis (HC vs. AD) and maternal instructions (neutral vs. negative), using the factor Involvement from the TCS for mothers as the dependent variable. For Hypothesis 3, moderation was tested using a regression model with child ER (EmReg score) predicted by Involvement, child emotional reactivity (self-reported negative emotions during the tangram task), and their interaction. The PROCESS tool was used for moderation analysis. To explore the impact of instructions on child behavior, an ANOVA was conducted with diagnosis (AD vs. HC) and instructions (neutral vs. negative) as independent variables and child distress as the dependent variable.

## 3. Results

### 3.1. Manipulation Check

The means and standard deviations and the statistics for the mixed ANOVA for each emotion reported by mothers before and during the tangram puzzle for each condition are shown in Table 2. A significant effect of time was observed for anger, disappointment, shame, and relaxation. The first three increased from the baseline measurement to after the mother–child interaction, whereas the last decreased, showing a stress induction. There was no significant interaction effect of condition and time point for any emotion (see Table 2). Also, there was no significant effect of condition on the emotions. In the questionnaire at the end of the experimental session to judge the performance of the computer game, mothers in the neutral condition reported being significantly more satisfied with the performance of their child during the computer puzzle game than mothers in the negative condition: *M*neutral = 8.96 (*SD* = 2.04), *M*negative = 7.66 (*SD* = 2.52), *t*(127) = −1.66, *p* = 0.01, *d* = 0.41.

### 3.2. Hypothesis 1: Child ER

Tests of normality (Kolmogorov–Smirnov and Shapiro–Wilk) were significant for both groups (all *p* < 0.05), indicating that the assumption of normality was violated. Nevertheless, the ANOVA was conducted because it is robust to moderate violations of normality, particularly when group sizes are approximately equal and the sample size is sufficiently large (*n* > 30 per group; Schmider et al. [66]). Moreover, Levene’s test indicated homogeneity of variances (*p* = 0.182), suggesting that the main assumptions for ANOVA were largely met. The EmReg score was significant higher for children in the HC group than for children in the AD group, *F*(1, 127) = 10.67, *p* ≤ 0.001, *ηp*^2^ = 0.08, thus showing higher ER skills in HC children than in children with ADs. Means and standard deviations are shown in Table 3.

### 3.3. Hypothesis 2: Maternal Involvement and Expectations

Again, tests of normality (Kolmogorov–Smirnov and Shapiro–Wilk) were significant for all groups (all *p* < 0.05), indicating that the assumption of normality was violated. Nevertheless, the ANOVA was conducted because it is robust to moderate violations of normality, particularly when group sizes are approximately equal and the sample size is sufficiently large (*n* > 30 per group; Schmider et al., [66]. Moreover, Levene’s test indicated homogeneity of variances (*p* = 0.221), suggesting that the main assumptions for ANOVA were largely met. Maternal involvement did not differ between the groups (AD vs. HC), *F*(1, 125) = 0.08, *p* = 0.779, *ηp*^2^ = 0.01 or the conditions (neutral vs. negative), *F*(1, 125) = 0.78, *p* = 0.381, *ηp*^2^ = 0.01. Further, the interaction term Group × Condition did not reach significance, *F*(1, 125) = 0.37, *p* = 0.55, *ηp*^2^ = 0.00. Thus, mother’s involvement did not depend on child anxiety diagnosis or on their expectations concerning the child’s struggle in the task. Means and standard deviations are shown in Table 3 and Table 4.

### 3.4. Hypothesis 3: Prediction of Child ER

A moderation analysis was run to determine if the interaction between mother’s involvement and emotional reactivity of the child significantly predicts child ER. As there were significant deviations from normality in the variable involvement, a bootstrapping procedure with 5000 samples was used in the moderated regressions. The relationship of all variables involved in the moderation analysis was approximately linear, as assessed by visual inspection of the scatterplots after locally estimated scatterplot smoothing. The overall model was significant, *F*(3, 124) = 8.70, *p* ≤ 0.001, predicting 17.39% of the variance.

Table 5 shows the results of the moderated regression with the mean-centered predictor and moderator. Emotional reactivity moderated the effect between involvement and EmReg score significantly, *ΔR^2^* = 0.04%, *F*(1, 123) = 5.90, *p* = 0.019. Figure 2 shows the simple slope analyses of the conditional effect of involvement on ER at three levels (mean value and ±1 *SD*) of emotional reactivity. The negative association between involvement and EmReg score was of greater magnitude in participants with greater emotional reactivity (i.e., 1 *SD* above the mean, *b* = −0.48, confidence interval 95% [CI] [−0.07, −0.03]) compared to those with lower emotional reactivity (i.e., at the mean, *b* = −0.03, 95% CI [−0.05, −0.01], or 1 *SD* below the mean, *b* = −0.01, 95% CI [−0.32, 0.18]). The Johnson–Neyman technique indicated that for mean centered emotional reactivity levels at or above −42.00 (57.01% of our sample) the relationship between involvement and ER skills was negative and significant. For emotional reactivity below this score (42.97% of our sample), there was a nonsignificant relationship between the measures. Only for children experiencing high emotional reactivity in the experimental situation was the relation between involvement and ER significant.

### 3.5. Exploratory Analysis: Effects of Expectations on Child Behavior

Since the tests of normality (Kolmogorov–Smirnov and Shapiro–Wilk) were significant for both groups (all *p* < 0.05), the assumption of normality was not met. However, ANOVA was conducted because it is robust against moderate violations of this assumption, especially when group sizes are comparable and each group includes a sufficiently large number of participants (*n* > 30; Schmider et al., [66]). Again, Levene’s test indicated homogeneity of variances (*p* = 0.194), suggesting that the main assumptions for ANOVA were largely met. Children in the AD group showed more distressed behavior than children in the HC group, *F*(1, 125) = 8.49, *p* = 0.004, *ηp*^2^ = 0.06. Maternal expectations did not have an influence on child distress, *F*(1, 125) = 0.56, *p* = 0.457, *ηp*^2^ = 0.004. Still, group and condition did not show a significant interaction effect, *F*(1, 125) = 2.42, *p* = 0.122, *ηp*^2^ = 0.02. Means and standard deviations are shown in Table 3 and Table 4.

## 4. Discussion

This study aimed to shed light on several maintaining factors of ADs, such as ER and emotional reactivity in children, maternal overinvolvement and expectations concerning their child’s ability, and child distress. This exploration was conducted through an observational design, allowing for the examination of the factors in a situational context. As expected, we found less ER ability in children with ADs compared to HC children. Contrary to our assumption, mothers of children with ADs did not differ from mothers of HC children in observed overinvolvement while their child solved a difficult tangram task. Overinvolvement was not strengthened by experimentally induced negative expectations concerning the child’s ability. Consistent with our hypothesis, results suggest that emotional reactivity moderated the relationship between overinvolvement and ER abilities. Specifically, we found that for children experiencing high emotional reactivity in the experimental situation, the relationship between overinvolvement and ER dysregulation became significant and stronger.

In our exploratory analysis, we found that children with ADs showed more distressed behavior than HC children. The expectation of mothers concerning their child’s ability did not have an effect on the observed child distress.

### 4.1. ER and ADs

In line with previous research [16,59], children with ADs showed lower ER abilities compared to HC children. To the best of our knowledge, this study represents the first instance of employing an observational measure to confirm this effect. It indicates that children with ADs not only report perceived ER problems but also exhibit observable deficits in regulating emotions. Notably, our study revealed deficits in ER in a context primarily associated with stress rather than anxiety. This is in line with the results of Suveg and Zeman [12], who reported that deficits appear to extend beyond situations that induce anxiety or sadness and encompass other negative emotions, such as stress. This outcome once again underscores the role of ER as an underlying factor contributing to the persistence of various anxiety disorders [11,17]. Furthermore, our findings suggest that situational, observable ER during the Tangram task is also impaired in children with anxiety disorders, not just the long-term, trait-based ER typically assessed via questionnaires. This also extends prior work (e.g., [16]) by showing that situational ER deficits coexist with—or even precede—trait-based dysregulation in anxious children. Situational ER deficits may be more amenable to intervention (e.g., real-time coaching during stressful tasks) compared to trait-based deficits, which require longer-term skill-building.

### 4.2. Maternal Overinvolvement of HC Children and Children with ADs

Contrary to our assumption and previous findings [21,22,25,42], mothers of children with ADs did not exhibit differences in observed overinvolvement. Notably, in our sample, 30% had a specific phobia as the primary diagnosis, in contrast to the 9% reported by Hudson and Rapee [22]. It has been suggested that specific phobias especially in childhood overlap less with other ADs and tend to be relatively more “pure” than other ADs [67]. This implies that childhood phobias might have different underlying mechanisms or less impact on daily life compared to other ADs. In line with other studies on ADs (e.g., [21,22] and Gar and Hudson [21]), we included all ADs. Future research should further differentiate between the different ADs.

During our data collection, coinciding with and following the COVID-19 pandemic, anxiety prevalence surged [10]. While parental influence is often considered a key factor in the development and maintenance of AD [10], the pandemic may have shifted this dynamic. Disruptions to social environments and critical life events—such as school closures, isolation, or family stress—likely played a more dominant role in sustaining or exacerbating anxiety symptoms during this period [10,68,69,70]. Despite these contextual changes, our cross-sectional analysis revealed no significant differences in psychopathological outcomes (e.g., anxiety levels) among children when comparing pre- and post-pandemic groups. However, the cross-sectional design limits our ability to draw causal conclusions. Long-term observation is needed to understand behavioral changes following major events such as the pandemic.

As noted by van der Bruggen et al. [43], the relationship between ADs and parental overcontrol lacks clarity. Focusing solely on parental behavior may offer a limited view of the complex parent–child relationship dynamics. To gain a deeper understanding, it would be valuable to consider additional dimensions such as communication patterns and child behavior in studying this dynamic interplay. In the current findings, a potential explanation for the observed results could be gender differences. Krohne and Hock [71] reported distinctions exclusively among girls and not among boys. However, because of limitations in sample size, incorporating gender as an additional factor in the current analysis was unfeasible. A larger sample or alternative methods could reveal more nuanced gender-related distinctions, warranting further investigation.

The methodological distinctions between our study and the work of Hudson and Rapee [22] introduce important considerations for interpretation. In their study, mothers received a solution booklet, whereas in ours, they did not. This difference suggests that observed maternal involvement levels in our study could have been linked to mothers’ ability to offer suitable assistance, a variable not systematically controlled for. Future research should include measures to assess this, clarifying potential correlations between maternal involvement and problem-solving effectiveness. Furthermore, variations in the psychopathological profiles of mothers across the different groups introduce an additional layer of complexity [72]. The interplay between maternal psychopathology and observed behaviors merits closer examination, as it has the potential to significantly impact the dynamics within parent–child interactions [73,74]. For instance, Woodruff-Borden et al. [73] found that mothers with ADs tended to be more withdrawn and avoided negative emotions in their children. Future studies would benefit from integrating this inquiry into their design, deepening our understanding of parent–child interactions and potentially shaping these interactions.

### 4.3. The Role of Maternal Expectation in Involvement

No differences were found concerning the different expectations of mothers in their involvement. The effectiveness of the instructional manipulation remains unclear, as emotional ratings of mothers in neutral and negative conditions showed no disparity. Only the rating of how satisfied mothers were with the performance of their child differed between the conditions. However, methodological distinctions between our study and Creswell et al. [46] are noteworthy. Creswell et al. [46] included positively formulated instructions for parents, whereas we used neutral instructions. Additionally, in Creswell et al.’s [46] study, parents were allowed to help as they saw fit, whereas in our study, mothers were instructed to intervene only if necessary. This instruction may have heightened maternal attentiveness to genuine child needs, possibly resulting in a reluctance to intervene despite a desire to assist, particularly for mothers under negative instructions. The study conducted by Asbrand et al. [27] utilized similar instructions to those in our study and identified distinctions in maternal overinvolvement. Notably, their investigation exclusively involved children with social anxiety disorder, excluding those with other forms of ADs. In future research expectations should be manipulated in a stronger way. A specific manipulation check should be included to explicitly assess participants’ expectations. Knowing more about the expectations of mothers toward children with ADs compared to mothers of HC children could be helpful for developing intervention programs. Qualitative studies in which mothers are asked about their expectations could be particularly informative.

### 4.4. The Role of Emotional Reactivity, Maternal Overinvolvement, and Child ER

As expected, emotional reactivity moderated the relationship between overinvolvement and ER, becoming stronger for children experiencing higher emotional reactivity in the experimental situation. This result indicates that children perceiving more emotionality are more vulnerable to overinvolved parenting and this influences their ability to regulate their emotions. The situational analysis of the problems of ER might extend in longer terms to problems in adjustment specially to internalizing problems. This would be in line with the finding of Morris et al. [41], in which mothers’ control was associated with internalizing problems for children with high irritability. Also, Allen et al. [34] propose that parental behaviors (e.g., overcontrol, anxious modeling) disproportionately affect vulnerable children, disrupting their cognitive and emotional processing (e.g., threat appraisal, emotion regulation). These early disruptions act as precursors to internalizing problems (e.g., ADs), particularly during key developmental phases. Over time, poor ER may thus lead to broader adjustment difficulties, reinforcing a cycle of maladaptive responses. The direction of this effect remains unclear; it would also be important to examine whether the self-reported higher emotionality can be observed physiologically—for example, through altered heart rate variability. Future research using a longitudinal design is needed to clarify this. Our finding highlights the importance of the interaction between parenting behavior and child characteristics, as already outlined by Morris et al. [41], Belsky et al. [30] and Allen et al. [34]. Focusing solely on differences in parenting behavior may indeed offer an incomplete understanding of the intricate dynamics at play. To comprehensively grasp the complexities involved, researchers should consider adopting a more holistic approach.

### 4.5. Exploratory Analysis

In our exploratory analysis, no difference in child distress was found depending on the instructions mothers received. As mentioned earlier, the manipulation of the instructions may not have been strong enough for us to detect an effect. In future studies, adjustments to the instructions could be considered to enhance the likelihood of identifying significant effects. In any case, varying levels of child distress could be observed, with the group of children with ADs exhibiting more child distress compared to the HC group. This is in line with Asbrand et al. [27] who also found the tendency of more distressed behavior in children with social phobia. Also, Dumas et al. [74] showed that the interaction between mothers and children with ADs was more negative, with children also showing more negative affect. As mentioned above, maternal overinvolvement did not differ between groups in our study. Also, Asbrand et al. [27] found that child distress predicts maternal negativity rather than overinvolvement. In this context, conducting additional analyses, including of parental negativity, to understand the relationship between parenting behavior and child behavior could be interesting.

### 4.6. Limitations and Strengths

Several limitations and some strengths warrant careful consideration in the interpretation of our findings. First, it is imperative to acknowledge that only satisfactory and not excellent statistical interrater reliability was achieved for child distress and the EmReg score. Consequently, prudence is advised when interpreting the results. Nevertheless, multiple blinded coders were involved, assessing the entire data set. This can be considered a strength, enhancing the reliability and objectivity of the evaluations, as it reduces the potential for bias in the analysis process. Additionally, for the most surprising finding concerning maternal involvement, interrater reliability was excellent. Second, the EmReg score, derived from a single item, provides a limited representation. To ensure coherence and comparability with prior research, we adopted the scoring system used by Hagstrøm et al. [55]. This decision enhances the clarity of our results and enables meaningful comparisons with previous studies.

Our reliance on studies predominantly using self-report for emotional reactivity is evident [38,59,75]. Solely using self-report measures for emotional reactivity assessment is limiting, so future research could bolster this by incorporating physiological measurements [76]. Nonetheless, our study’s strength lies in distinguishing between ER and reactivity, offering a nuanced understanding and enhancing the precision of our findings. A limitation is the extended data collection period (2016–2022), during which unmeasured societal events (e.g., the COVID-19 pandemic) may have influenced outcomes.

The study’s cross-sectional design impedes causal inferences, emphasizing the need for longitudinal research to clarify the directional influence between variables. Consequently, we can only analyze the observed associations without being able to determine the direction of the effects or their alignment with theoretical assumptions. The sample’s homogeneity, mainly white middle-class families, hinders generalizability to more diverse samples but reduces confounding variables. Focusing solely on mothers also minimizes heterogeneity in results and potential confounding influences. Furthermore, reactivity to the laboratory setting and the awareness that their behavior was under observation might have influenced parental and child behavior.

One particular strength of the study is the diagnostic clarity of the groups, and the adequacy of the sample size contributes to the robustness of our study. Furthermore, the inclusion of non-self-report measures and the implementation of an observational design with experimental manipulation enhances the methodological rigor of our investigation.

## 5. Conclusions

Our study underscores the importance of child ER for the maintenance of ADs. We demonstrated this effect by utilizing an observational design, rather than relying solely on self-report questionnaires. Although ER in our study is considered as an overall construct, our findings have the potential to complement existing research, particularly studies such as that of Schäfer et al. [16]. In their work, they identified psychopathology-specific outcomes related to anxiety symptoms. It could be valuable to integrate ER training into early intervention programs or into therapy programs to treat ADs in a more effective way [17,77]. Interestingly, mothers’ involvement did not differ between our experimental groups and was not influenced by their expectations concerning their child’s abilities. But the interaction between overinvolvement and emotional reactivity influenced ER. Investigating how maternal behavior interacts with child factors such as emotional reactivity and ER could shed light on AD maintenance. Future studies should consider incorporating additional factors like child characteristics (e.g., temperament, executive function) or contextual stressors, e.g., critical life events, to better understand the complex relationship among parenting behavior, child characteristics, critical life events, and ADs. It could be interesting to adopt ecological momentary assessment (EMA) and advanced interaction measurement methods to capture the dynamic, bidirectional nature of these processes [78]. For instance, EMA could reveal how maternal overinvolvement fluctuates in response to real-time child anxiety, while automated behavioral analysis (e.g., OpenFace) might detect subtle nonverbal cues that manual coding misses ([79]). Future research should also incorporate longitudinal studies. While cross-sectional and experimental studies (including our own) provide critical insights into associations and short-term dynamics, only longitudinal research can elucidate causal pathways, developmental trajectories, and sensitive periods in which parenting behaviors exert their strongest effects on child anxiety.

## Figures and Tables

**Figure 1 ijerph-22-01802-f001:**
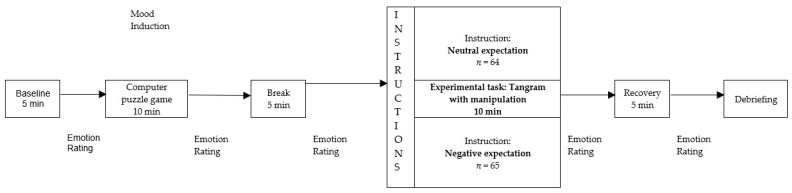
Procedure of the Laboratory Session.

**Figure 2 ijerph-22-01802-f002:**
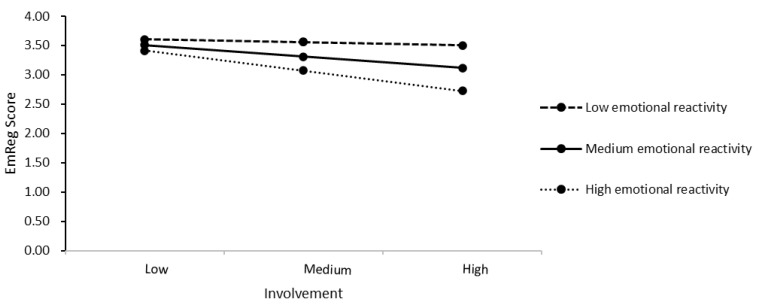
Simple Slopes Analyses. Note. Graph showing the relationship between involvement and ER in children for different levels (mean value and ±1 *SD*) of emotional reactivity. EmReg = the overall ER scale.

**Table 1 ijerph-22-01802-t001:** Participant Characteristic of the Anxiety Disorder (AD) and Healthy Control (HC) Groups.

Variable	AD (*N* = 64)	HC (*N* = 65)	Statistics
Child’s age (in years), *M* (*SD*)	11.14 (1.5)	11.25 (1.67)	*t*(127) = 0.40, n.s.
Mother’s age ^a^ (in years), *M* (*SD*)	44.56 (4.67)	43.67 (4.73)	*t*(124) = −0.06, n.s.
Female, *n* (%)	35 (55%)	36 (55%)	χ^2^(1) = 0.06, n.s.
Elementary school/secondary school	15/32	15/34	χ^2^(1) = 0.03, n.s.
SCAS mother report ^a^	22.56 (11.08)	6.50 (4.83)	*t*(124) = −3.20 ***
SCAS child report ^b^	27.74 (16.06)	10.21 (7.64)	*t*(107) = −7.30 ***
SCL-27 ^a^	0.35 (0.29)	0.21 (0.18)	*t*(124) = −3.2 ***
CDI ^c^	12.13 (7.52)	5.43 (4.39)	*t*(106) = −5.99 ***

Note. Percentages do not always add up to 100 because of rounding. CDI = Child Depression Inventory [49]; SCAS = Spence Children’s Anxiety Scale [50,51]; SCL-27 = Symptom Checklist 27 [52]. ^a^ Missing data: *n*_HC_ = 1, *n*_AD_ = 3. ^b^ Missing data: *n*_HC_ = 10, *n*_AD_ = 10. ^c^ Missing data: *n*_HC_ = 10, *n*_AD_ = 11. *** *p* ≤ 0.001, n.s. = not significant.

**Table 2 ijerph-22-01802-t002:** Means, Standard Deviations and Mixed-Model ANOVA Statistics for the Emotion Ratings of the Mothers Before and After the Tangram Puzzle.

Emotion	Condition						ANOVA			
	Neutral			Negative						
	*N*	*M*	*SD*	*N*	*M*	*SD*	Effect	*F* Ratio	*df*	*η* ^2^
Anxiety							C	0.3	1, 125	0.00
Before tangram	63	2.13	11.10	64	1.42	1.77	T	1.91	1, 125	0.02
After tangram	63	2.25	5.93	64	4.22	11.92	C × T	1.59	1, 125	0.01
Anger							C	0.59	1, 124	0.36
Before tangram	62	1.08	2.54	64	2.70	9.63	T	50.53 ***	1, 124	0.29
After tangram	62	15.53	19.44	64	17.26	25.27	C × T	0.001	1, 124	0.00
Disappointment							C	0.49	1, 124	0.00
Before tangram	62	2.03	5.21	64	3.42	11.45	T	83.9 ***	1, 124	0.40
After tangram	62	22.54	26.5	64	25.09	28.39	C × T	0.06.	1, 124	0.00
Relaxation							C	0.25	1, 124	0.91
Before tangram	62	75.17	22.13	64	78.99	24.1	T	110.81 ***	1, 124	0.47
After tangram	62	50.61	26.78	64	43.33	27.06	C × T	3.77	1, 124	0.03
Happiness							C	1.91	1, 123	0.02
Before tangram	62	52.73	26.27	63	50.63	27.10	T	2.00	1, 123	0.02
After tangram	62	44.62	28.37	63	50.54	26.14	C × T	1.91	1, 123	0.02
Pride							C	1.42	1, 124	0.01
Before tangram	62	21.18	26.20	64	18.13	25.71	T	2.36	1, 124	0.02
After tangram	62	26.19	25.76	64	20.07	22.51	C × T	0.46	1, 124	0.00
Shame							C	0.70	1, 124	0.01
Before tangram	62	0.98	3.05	64	2.78	9.76	T	20.39 ***	1, 124	0.14
After tangram	62	8.05	15.89	64	9.30	18.49	C × T	0.03	1, 124	0.00
Sadness							C	7.56	1, 96	0.07
Before tangram	50	1.06	2.34	48	7.27	16.54	T	3.67	1, 96	0.04
After tangram	50	4.48	10.29	48	10.97	22.12	C × T	0.01	1, 96	0.00

Note. ANOVA = analysis of variance; C = condition; Neutral= neutral instructions; Negative = negative instructions; T = Time. *** *p* < 0.001.

**Table 3 ijerph-22-01802-t003:** Means, Standard Deviations and Statistics Depending on Diagnostic Status.

Variable	Group		Statistics
	AD	HC	
	*M* (*SD*)	*M* (*SD*)	
Maternal involvement	4.00 (1.3)	4.01 (1.35)	*F*(1, 125) = 0.08, n.s.
EmReg score	3.12 (0.78)	3.54 (0.67)	*F*(1, 127) = 10.66 **
Child distress	2.62 (0.99)	2.38 (0.82)	*F*(1, 125) = 8.49 **

Note. AD = Anxiety disorder; HC = healthy control. EmReg = the overall ER scale. ** *p* < 0.01. n.s. = Not significant.

**Table 4 ijerph-22-01802-t004:** Means and Standard Deviations and Statistics Depending on Experimental Condition.

Variable	Condition		Statistics
	Neutral	Negative	
	*M* (*SD*)	*M* (*SD*)	
Maternal involvement	4.12 (1.3)	3.93 (1.54)	*F*(1, 125) = 0.78, n.s.
Child distress	2.69 (1.05)	2.56 (0.92)	*F*(1, 125) = 0.56, n.s.

Note. n.s.= not significant.

**Table 5 ijerph-22-01802-t005:** Moderated Regression.

Predictor	*b*	95% CI	*SE*	*t*	*p*
Constant	3.31	[3.19, 3.43]	0.06	53.33	≤0.001
Involvement	−0.03	[−0.04, −0.01]	0.01	−3.13	≤0.001
Emotional reactivity	−0.002	[−0.00, −0.00]	0.00	−3.85	≤0.001
Involvement × Emotional Reactivity	−0.0002	[−0.03, 0.00]	0.00	−2.43	0.010

Note. *N =* 128. *F*(1, 124) = 5.90. *p* = 0.017, *ΔR*^2^ = 0.04. CI = Confidence interval.

## Data Availability

The data that support the findings of this study are available from the corresponding author upon reasonable request. The data are not publicly available due to ethical and legal restrictions to protect the privacy of the participating children and their families.

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
