# Peer review of "Maternal Expectations and Overinvolvement, and Child Emotion Regulation in Children with and Without Anxiety Disorders: An Experimental Observational Study"

_ijerph, 2025, doi:10.3390/ijerph22121802_

Round 1
Reviewer 1 Report
Comments and Suggestions for Authors
Dear Author(s),
I congratulate you on your efforts up to the review stage of your manuscript. The title of this manuscript is “Maternal expectations and overinvolvement, and child emotion regulation in children with and without anxiety disorders: An experimental observational study.” I have shared the strengths and improvable aspects of this research below.
Strengths:
- The introduction section goes straight to the point. It is written in a fluent manner.
- A study has been completed on an important topic, as anxiety problems are more common today.
- The elements that distinguish this research from other studies in the literature are sufficiently explained.
- Ethical committee approval has been obtained.
- The data analysis of the research is explained in detail.
- Data analyses are explained in detail.
- There are a sufficient number of references.
- It is explained which references the comments are based on.
- The discussion section is strong. The limitations of the observations and comments are explained.
Weaknesses or questions to be answered:
- The introduction section includes “recommendations” and “predictions.” Authors should evaluate their findings and make recommendations in the discussion section. The introduction section requires revisions. The introduction section should not contain characteristics of the discussion section.
- The research approach and paradigm should be explained in detail in the methods section.
- As data was collected between 2016 and 2022, the research results may have been influenced by many factors. This should be clearly stated in the limitations.
- Permissions for the use of data collection tools should be specified.
- The content and duration of the videos should be explained.
- The research recommendations could be expanded.
- More information about the measurement tools should be provided to the reader.
- Although the data was collected between 2016 and 2022, the research references are quite old. For example, there are no references from 2025. One reference from 2024 and two references from 2023 were used. The research needs to be re-discussed with current references.
- An introduction to and discussion of theories on the mother-child relationship should be included.
Reviewer 2 Report
Comments and Suggestions for Authors
The manuscript addresses an important topic in developmental clinical psychology: the links between maternal expectations/overinvolvement and children’s emotion regulation in samples with and without anxiety disorders. The study is generally well grounded, methodologically clear, and the procedures are well described. The manuscript is overall suitable for publication pending minor revisions.
Consult attached file.

Round 2
Reviewer 1 Report
Comments and Suggestions for Authors
The authors carefully reviewed the referee feedback and made revisions. I believe the revisions have improved the article. I recommend publishing this work, which will contribute to the field. I thank the authors.